# Emergence of Nontuberculous Mycobacteria at the Human–Livestock–Environment Interface in Zambia

**Mildred Zulu** [1,2,3,*] 🆔 **, Sydney Malama** [3,4] **, Ngula Monde** [2,3,5] **, Henson Kainga** [3,6] **, Rebecca Tembo** [1,3] **, Florence Mwaba** [1,3] **, Shereen Ahmed Saad** [3,7] **, Victor Daka** [3,8] **, Andrew N. Mukubesa** [2] **, Joseph Ndebe** [2] **, Obi Shambaba** [5] **and Musso Munyeme** [2,3] 🆔

1   Department of Pathology and Microbiology, School of Medicine, University of Zambia, Lusaka P.O. Box 50110, Zambia
2   Department of Disease Control, School of Veterinary Medicine, University of Zambia, Lusaka P.O. Box 32379, Zambia
3   Africa Center of Excellence for Infectious Diseases of Humans and Animals, University of Zambia, Lusaka P.O. Box 32379, Zambia
4   Department of Biological Sciences, School of Natural Science, University of Zambia, Lusaka P.O. Box 32379, Zambia
5   Department of Biomedical Sciences, Tropical Diseases Research Centre, Ndola P.O. Box 71769, Zambia
6   Department of Veterinary Epidemiology and Public Health, Faculty of Veterinary Medicine, Lilongwe University of Agriculture and Natural Resources, Lilongwe 207203, Malawi
7   Department of Clinical Studies, College of Veterinary Science, University of Bahr El Ghazal, Wau P.O. Box 10739, South Sudan
8   Department of Public Health, School of Medicine, Copperbelt University, Ndola P.O. Box 71191, Zambia
*   Correspondence: mildredzulu@ymail.com; Tel.: +26-097-8162-748

**Abstract:** The prevalence of nontuberculous mycobacteria (NTM) infections and disease is rising worldwide due to increased research, diagnostics capabilities, and awareness of the disease. There is limited prevalence data for NTM from different sources in Zambia. The aim of this study was to determine the prevalence and species distribution of NTM at the human–livestock–environment interface. A cross-section study was conducted in Namwala, Chipata, and Lundazi Districts of Zambia from April 2020 to December 2021. Sputum samples were collected from tuberculosis presumptive patients from different health centers, cattle tissues were collected from different abattoirs during routine post-mortem, and water samples were collected from different drinking points for humans and animals such as taps, boreholes, wells, rivers, dams and ponds, and then cultured following standard mycobacteriology procedures. Capilia TB-Neo assay was used to identify NTM from the positive cultures. DNA was extracted and the 16S to 23S rRNA (internal transcribed spacer region) (ITS) was amplified and sequenced to identify the species. The overall prevalence of NTM from humans, cattle, and water was 9.1% (72/794, 95% CI 7.2–11.3). The prevalence in humans was 7.8% (33/421, 95% CI 5.54–10.94), in cattle it was 10.6% (15/142, 95% CI 6.2–17.1), and in water it was 10.4% (24/231, 95% CI 6.9–15.2). Our study has shown, for the first time in Zambia, simultaneous isolation of NTM at the human–livestock–environment interface; *M. avium* complex and *M. fortuitum* were the most commonly isolated species. *M. fortuitum* and *M. gordonae* were isolated from all three sources, while *M. abscessus* was isolated from humans and water. The isolation of similar NTM species at the interface which are potentially pathogenic is a public health problem which merits further investigation.

**Keywords:** cattle; nontuberculous mycobacteria; prevalence; water; human; interface

## 1. Introduction

The prevalence of nontuberculous mycobacteria (NTM) continues to rise worldwide with nontuberculous mycobacteria infection accounting for almost half of the total number of isolated mycobacteria [1–3]. The reasons for this increase are not clear, but they are

thought to be due to increased research into the epidemiology, diagnostics, and treatment of this once obscure disease which stems from the increasing numbers of cases being identified from populations with previously unknown and currently unidentified risk factors [4].

NTM are ubiquitous in the environment with the heaviest concentration found in soil (especially acidic or coastal soils) and water. They are associated with biofilm formation which contributes to disinfectants and antibiotic resistance despite their slow growth [4,5]. Soil and open water are the main sources of NTM which play major roles as the sources of human and animal infections [6].

Similar to tuberculosis (TB), NTM can occur throughout the body but pulmonary infection, lymphadenitis and skin and soft tissue infections are the most common sites in humans, although pulmonary is the most common site [7–9]. These organisms occupy environmental habitats that are shared between humans and animals, which are thought to be the major sources of disease acquisition, especially in engineered environmental habitats such as water distribution systems (WDS) where overlapping of human and mycobacteria inhabitants permit recurrent exposure [10]. Humans and animals can ingest or inhale NTM in water, aerosols, or dust. Inhaled aerosols appears to be the primary transmission route of NTM pulmonary disease (NTMPD), although possible human–human transmission of *Mycobacterium abscessus* has been described [11,12]. Mycobacteria aerosolize more readily than other bacteria because of their hydrophobic cell walls [9].

The prevalence of NTM varies among continents, regions, and countries. The true global burden of the disease is unknown and estimates are subject to under- and/or overestimation [1]. In Africa and the Middle East, the prevalence of NTM ranges from 4 to 15% among suspected TB cases and from 18 to 20% among suspected multi-drug resistant TB (MDR TB) cases [8,13]. In Sub-Saharan Africa with a higher burden of HIV, the prevalence of NTM ranges from 3.2 to 56%, although there is insufficient data due to lack of laboratory infrastructure [14–20]. Effective surveillance networks for TB exist in many countries including Zambia; however, they are underutilized for NTM detection and management [21]. This challenge is compounded by a lack of stringent regulations on reporting NTM-suspected cases to the public health department in most countries [8,22,23].

According to the American Thoracic Society (ATS) and the Infectious Diseases Society of America (IDSA), a diagnosis of NTM disease in humans requires the presence of clinical symptoms with the appropriate exclusion of other diseases, radiographic abnormalities, and microbiological culture (positive culture from two sputum samples)or positive culture from one bronchial wash or lavage) [24]. For clinicians, there are at least three factors that can help to differentiate between mycobacterial disease and colonization, i.e., the bacteria load, the species isolated, and whether or not there is clinical or radiographic progression of the disease [25].

Infections due to NTM in Zambia continue to be neglected both in humans and in animals, although high combined prevalences of 24.39% and 14.81% for humans and cattle, respectively, have been reported [26]. This rise in the number of NTM isolation is a public health concern as these organisms are both difficult to diagnose and to treat. Most NTM are naturally resistant to commonly used anti-TB drugs and the course of treatment for NTM is very long and the cost is high [27]; hence, knowing the prevalence of NTM from different sources is important as it will give a clear picture of the burden of NTM at the human–livestock–environment interface [28]. Therefore, in this study, we aimed to determine the prevalence and species distribution of NTM at the human–livestock–environment interface in Eastern and Southern provinces of Zambia.

## 2. Materials and Methods

### 2.1. Study Site and Design

A cross-sectional study was conducted in Namwala, Chipata, and Lundazi Districts of Zambia (Figure 1) from April 2020 to December 2021. The sites were purposively selected based on the known fact that the selected sites were among the districts with the highest concentrations of the human–livestock interface [29].

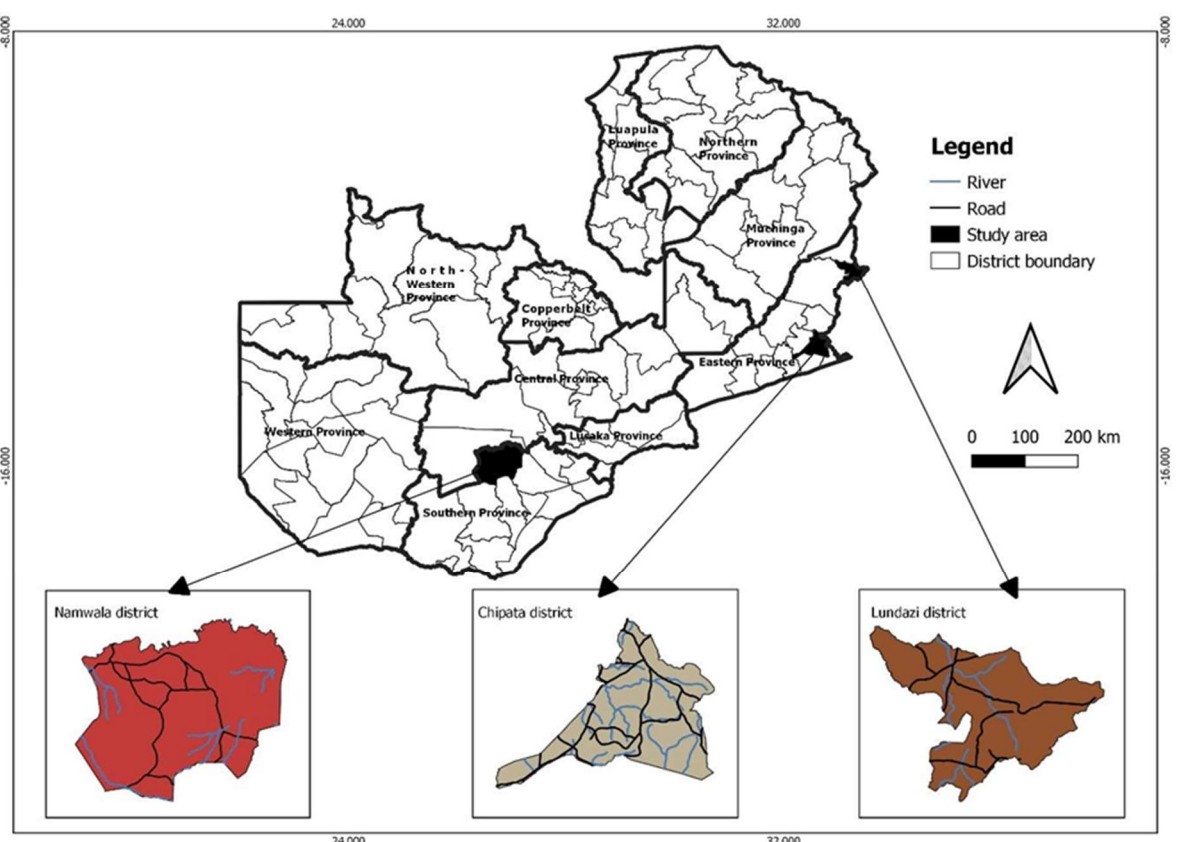

**Figure 1.** Map showing the study area of Namwala, Chipata and Lundazi Districts. Adopted from [30].

*2.2. Participants Recruitment and Sample Collection*

2.2.1. Human Sputum Samples

At each district health facility, tuberculosis presumptive patients (TPPs) were randomly selected from the patient register book, by selecting every third TPP; this was done during normal hospital operating hours. The TPPs were those patients who had had a cough for more than two weeks, and reported loss of appetite and night sweats. Sputum samples of about 2–10 mL were collected in a sterile sputum container with a tight-fitting lid. Then, the samples were transported under cold chain to the Tropical Diseases Research Centre (TDRC) laboratory, which is a level 3 Biosafety (BSL 3) TB regional reference laboratory in Ndola, where they were cultured immediately.

2.2.2. Cattle Tissues

The target population for the cattle tissues was comprised of cattle carcasses showing gross tubercle-like lesions at routine meat inspection at the abattoir. The cattle slaughtered at the abattoir were drawn from the villages where patients had come from seeking medical attention. The carcasses were examined according to the standard post-mortem abattoir examination procedure [31]. The specimens were collected using the purposive sampling method during routine abattoir work. TB-like lesions were collected in clean, sterile containers to avoid contamination with other environmental mycobacteria. Then, the samples were stored in a cooler box containing ice packs at 4 °C, and then transported to the TDRC for further analysis. A questionnaire which had information about the age, sex of the animal, herd size, and grazing systems among others was administered to the owners of the animals found with tubercle-like lesions.

### 2.2.3. Water Samples

Water samples were randomly collected from various drinking sources both for humans and for animals such as taps, wells, boreholes, ponds, streams, and rivers in the community of the study area. One hundred and fifty milliliters (150 mL) of water were collected in sterile whirlpak plastic bags. For taps and borehole sources, the water was allowed to run through for a while, and then was collected. Immediately after collection, all samples were transported to the TDRC where they were processed immediately. A bio data sheet was used to capture information such as the site of water collection, village name, and the purpose of water, among others.

### 2.3. Sample Processing

### 2.3.1. Sputum Samples

At the TDRC, which is a TB regional reference biosafety level 3 laboratory, the sputum samples were first decontaminated using the NALC-NAOH technique; 4% sodium hydroxide (NaOH) mixed with equal volumes of 2.9% sodium citrate solution and N-acetyllcysteine (NALC) was used as a working solution. An equal volume of NALC-NAOH working solution was added to each sputum sample and the samples were allowed to stand for 15 min. Thereafter, phosphate-buffered saline was added to the mixture up to the 45 mL mark. This was followed by centrifugation at 3000 rpm for 20 min. Then, the sediment was cultured using mycobacterial growth indicator tube (MGIT) (Becton, Dickinson and Company, East Rutherford, NJ, USA) liquid media. Positive cultures were examined microscopically for acid-fast bacillus using the Zielh Nelseen staining procedure to confirm the growth.

### 2.3.2. Tissue Sample

The tissue samples were examined macroscopically under a biosafety level II cabinet to remove small pieces of lesions by using a sterile scissors and forceps. Three (3) grams of the tissue were cut into small pieces using a sterile scalpel blade. Then, tissue samples were crushed in a clean mortar using a pestle, and then the minced tissue was transferred into a clean well-labelled centrifuge tube. Ten milliliters (10 mL) of 4% NaOH was added to the minced tissue for decontamination and allowed to stand for 15 min. This was followed by neutralization with PBS (PH 6.8) added up to the 50 mL mark, and the tube was centrifuged at $3000 \times g$ for 20 min at 4 °C. Then, the supernatant was carefully poured off leaving the sediments/pellet in the centrifuge tubes. Two (2) mL of PBS was added to the sediment, and then it was mixed well. A half milliliter (0.5 mL) of the sediment was inoculated onto two Lowenstein Jensen media both containing glycerol and incubated at 37 °C. Cultures were observed daily for the first week to identify fast growers, and then weekly for visible growth of bacteria until 12 weeks. Smears were prepared for each viable growth and Ziehl–Neelsen staining was performed to confirm the presence of AFB.

### 2.3.3. Water Sample Processing

At the TDRC, one hundred and fifty milliliters (150 mL) of water was filtered through 0.45 μL nitrocellulose membrane filters (Millipore Corporation, Bedford, MS, USA) by vacuum filtration using a Manifold Filtration System (Sartorius AG, Goettingen, Germany). Thereafter, the samples were further processed according to [32].

### 2.4. Identification of NTM

To identify NTM, Capilia TB-Neo assay (TAUNS Laboratories, Inc., Izunokuni, Japan) was used, according to the manufacturer's instructions. The test is an immuno-chromatographic-based method that detects the presence of MPB64 antigens specifically produced by MTBC and not by NTM. The results were interpreted after 15 min. Positive results suggested that the organism isolated was MTBC, and negative results indicated that it was NTM.

### 2.5. DNA Extraction and Amplification

DNA was extracted from the culture-positive isolates from sputum, cattle tissues, and water using a commercial genomic DNA extraction kit, Qiagen QlAamp DNA miniprep kit (Qiagen Group, Hilden, Germany), according to the manufacturer's instructions. The extracted DNA was amplified using one set of primers namely 16S–23S rRNA (ITS) genes using the following primers SP1 (5′-ACC TCC TTT CTA AGG AGC ACC-3′) and SP2 (5′-GAT GCT CGC AAC CAC TAT CCA-3′).

The PCR reaction was performed in a final reaction volume of 20 µL, consisting of 2 µL DNA template, 6 µL nuclease-free water, 10 µL One Taq Quick-Load (Biolabs, Durham, NC, USA), and 1 µL for each of the primers on a thermal cycler (Applied Biosystems, Chiba, Japan). The following PCR conditions were used: 95 °C for 5 min of initial denaturation, 40 cycles of 30 s at 95 °C, 30 s at 56 °C, 1 min at 72 °C, followed by a final extension period of 7 min at 72 °C. Electrophoresis was run at 100 V in 1.5% agarose (3.75 g in 250 mL of 1 × TAE buffer) (SigmaR) gel and visualized under UV light. During electrophoresis, Ethidium bromide was used as a staining reagent to visualize the PCR products alongside a 100 bp DNA ladder.

### 2.6. Purification of the PCR Products and Cycle Sequencing

The amplified PCR products were purified using the Zymo Research Genomic DNA Clean and Concentrator TM-25 kit (Irvine, CA, USA), according to the manufacturer's instructions. Then, the purified DNA products were sequenced based on sequencing of the 16S–23S ribosomal RNA genes using the following primers SP1 (5′-ACC TCC TTT CTA AGG AGC ACC-3′) and SP2 (5′-GAT GCT CGC AAC CAC TAT CCA-3′) and brilliant dye chain terminator ver. 3.1 (NimaGen, Nijmegen, The Netherlands). The sequence products were precipitated using the ethanol precipitation method followed by denaturation with formamide and capillary electrophoresis on an ABI 3500 Genetic Analyzer (Applied Biosystems, Chiba, Japan).

### 2.7. Data Analysis

Data were entered, cleaned, and validated in Microsoft™ Excel Spreadsheet (Microsoft Office Excel® 2019. Nontuberculous mycobacteria infection (positive or negative) was the dependable variable in this study, while gender of participants, age, province, district, water sources, flood plains, and grazing with animals were independent variables. Then, the data were exported to SPSS software ver. 21 (IBM Corp, Armonk, NY, USA). The descriptive results were frequencies and prevalence proportions which were presented in tables. The variables were significant at $p < 0.05$ and all tests were performed at a 95% confidence level.

The obtained sequences were assembled and edited using the ATGC plug-in Genetyx ver bioinformatics software, and then the sequences were subjected to blast analysis on the National Centre for Biotechnology Information (NCBI) website (https://blast.ncbi.nlm.nih.gov/Blast.cgi) accessed on 19 July 2022 for species identification.

## 3. Results

### 3.1. Prevalence of Nontuberculous Mycobacteria in Humans

A total of 421 participants were enrolled in the study. Of these, 231 (231/421) participants were from Namwala District, 109 (109/421) participants were from Chipata District, and 81 (81/421) participants were from Lundazi District. Male participants were the most common in Namwala (147) and Chipata (75) Districts, with the age group of 41–50 years being the most common age group in the two districts with (46) and (21) for Namwala and Chipata Districts, respectively, while female participants in the age group of 21–30 years were the most common in Lundazi District (43). The overall prevalence of NTM in humans was 7.8% (33/421, 95% CI 5.5–10.9), and male participants had a prevalence of 4.5% (19/421, 95% CI 2.8–7.1), which was not significantly different from that of female participants 3.3% (14/421, 95% CI 1.9–5.7) ($p = 0.899$).

Namwala District had a significantly higher overall prevalence of 4.5% (19/421, 95% CI 2.8–7.1) than Lundazi District 0.5% (2/421, 95% CI 0.1–1.9) (*p* = 0.019), while Chipata District had a higher district-specific prevalence of 11.0% (12/109, 95% CI 6.1–18.8) than Lundazi District 2.5% (2/81, 95% CI 4.3–9.5) (*p* = 0.022). The age group of 31–40 years had the highest overall and specific prevalences of 2.9% (12/421, 95% CI 1.6–5.1) and 10.7% (12/112, 95% CI 5.9–18.3), respectively (Table 1).

**Table 1.** Overall and specific prevalence of NTM in humans across age, sex, province, and districts.

| Variable | Category | Frequency | Prevalence (%) (*n* = 33) | 95% CI |
|---|---|---|---|---|
| Gender | Male | 421 | 19 (4.5) | 2.8–7.1 |
|  | Female | 421 | 14 (3.3) | 1.9–5.7 |
| *p*-value |  |  | 0.899 |  |
| Gender specific |  |  |  |  |
|  | Male | 252 | 19 (7.5) | 4.7–11.7 |
|  | Female | 169 | 14 (8.3) | 4.8–13.8 |
| *p*-value |  |  | 0.091 |  |
| Age (years) | ≤20 | 421 | 2 (0.5) | 0.1–1.9 |
|  | 21–30 | 421 | 6 (1.4) | 0.6–3.2 |
|  | 31–40 | 421 | 12 (2.9) | 1.6–5.1 |
|  | 41–50 | 421 | 9 (2.1) | 1.1–4.2 |
|  | 41–60 | 421 | 2 (0.5) | 0.1–1.9 |
|  | ≥61 | 421 | 2 (0.5) | 0.1–1.9 |
| *p*-value |  |  | 0.899 |  |
| Age specific |  |  |  |  |
|  | ≤20 | 22 | 2 (9.1) | 1.6–30.6 |
|  | 21–30 | 107 | 6 (5.6) | 2.3–12.3 |
|  | 31–40 | 112 | 12 (10.7) | 5.9–18.3 |
|  | 41–50 | 96 | 9 (9.4) | 4.6–17.5 |
|  | 41–60 | 42 | 2 (4.8) | 8.3–17.4 |
|  | ≥61 | 42 | 2 (4.8) | 8.3–17.4 |
| *p*-value |  |  | 0.677 |  |
| Province | Southern | 421 | 19 (4.5) | 2.8–7.1 |
|  | Eastern | 421 | 14 (3.3) | 1.9–5.7 |
| *p*-value |  |  | 0.899 |  |
| Province specific |  |  |  |  |
|  | Southern | 231 | 19 (8.2) | 5.5–12.7 |
|  | Eastern | 190 | 14 (7.4) | 4.2–12.3 |
| *p*-value |  |  | 0.091 |  |
| Districts | Namwala | 421 | 19 (4.5) | 2.8–7.1 |
|  | Chipata | 421 | 12 (2.9) | 1.6–5.1 |
|  | Lundazi | 421 | 2 (0.5) | 0.1–1.9 |
| *p*-value |  |  | 0.019 |  |
| District specific |  |  |  |  |
|  | Namwala | 231 | 19 (8.2) | 5.5–12.7 |
|  | Chipata | 109 | 12 (11.0) | 6.1–18.8 |
|  | Lundazi | 81 | 2 (2.5) | 4.3–9.5 |
| *p*-value |  |  | 0.022 |  |
|  | Overall | 33 | 7.8 | 5.5–10.9 |

*n*, number of participants; CI, 95% confidence interval.

### 3.2. Prevalence of NTM in Cattle

A total of 142 cattle tissues were collected, of which 129 (129/142) cattle were from Namwala District, 9 (9/142) cattle were from Chipata District, and 4 (4/142) cattle were from Lundazi District. The majority of the cattle from Namwala District were females (73/129), while for Chipata District, the majority of animals were males (6/9) and Lundazi District had equal numbers of animals sampled. The most common age group for the sampled animals was between the ages of 6 and 10 years for all the districts. The overall prevalence of NTM in cattle was 10.6% (15/142, 95% CI 6.2–17.1). Male animals had a

significantly high overall and specific prevalence of 7.0% (10/142, 95% CI 3.6–12.9) and 12.5% (10/80, 95% CI 6.5–22.2) than females, 3.5% (5/142, 95% CI 1.3–8.5), and 8.1% (5/62, 95% CI 3.0–18.5), ($p$ = 0.001) and ($p$ = 0.031), respectively. The age group from 6 to 10 years had the highest overall and category specific prevalence of 8.5% (12/142) (95% CI 4.6–14.6) and 12.1% (12/99, 95% CI 6.7–20.6), while that below five years did not record any positives for NTM infection (Table 2).

**Table 2.** Overall and specific prevalence of NTM infection across age, districts, and sex of the cattle.

| Factors | Category | Frequency | Prevalence (%) $n = 15$ | 95% CI |
|---|---|---|---|---|
| Age of livestock in years | ≤5 | 142 | 0 (0.0) | 0.0–3.3 |
| | 6–10 | 142 | 12 (8.5) | 4.6–14.6 |
| | ≥11 | 142 | 3 (2.1) | 0.6–6.5 |
| *p*-value | | | 0.047 | |
| Age specific | | | | |
| | ≤5 | 3 | 0 (0.0) | 0.0–69.0 |
| | 6–10 | 99 | 12 (12.1) | 6.7–20.6 |
| | ≥11 | 40 | 3 (7.5) | 2.0–21.5 |
| *p*-value | | | 0.021 | |
| District | Lundazi | 142 | 0 (0.0) | 0.0–3.3 |
| | Chipata | 142 | 0 (0.0) | 0.0–3.3 |
| | Namwala | 142 | 15 (10.6) | 6.2–17.1 |
| *p*-value | | | 0.007 | |
| District specific | | | | |
| | Lundazi | 4 | 0 (0.0) | 0.0–60.4 |
| | Chipata | 9 | 0 (0.0) | 0.0–37.1 |
| | Namwala | 129 | 15 (11.6) | 6.9–18.8 |
| *p*-value | | | 0.014 | |
| Sex | Male | 142 | 10 (7.0) | 3.6–12.9 |
| | Female | 142 | 5 (3.5) | 1.3–8.5 |
| *p*-value | | | 0.001 | |
| Sex specific | | | | |
| | Male | 80 | 10 (12.5) | 6.5–22.2 |
| | Female | 62 | 5 (8.1) | 3.0–18.5 |
| *p*-value | | | 0.031 | |
| Herd size | ≤50 | 142 | 3 (2.1) | 0.6–6.5 |
| | 51–150 | 142 | 8 (5.6) | 2.6–11.2 |
| | ≥151 | 142 | 4 (2.8) | 0.9–7.5 |
| *p*-value | | | 0.472 | |
| Herd specific | | | | |
| | ≤50 | 13 | 3 (3.1) | 6.1–54.0 |
| | 51–150 | 65 | 8 (12.3) | 5.8–23.4 |
| | ≥151 | 64 | 4 (6.3) | 2.0–16.0 |
| *p*-value | | | 0.001 | |
| Housed | Yes | 142 | 15 (10.6) | 6.2–17.1 |
| | No | 142 | 0 (0.0) | 0.00–3.3 |
| *p*-value | | | 0.001 | |
| Housed specific | | | | |
| | Yes | 138 | 15 (11.0) | 6.4–17.6 |
| | No | 4 | 0 (0.0) | 0.00–60.4 |
| *p*-value | | | 0.001 | |
| Flood plain | Yes | 142 | 15 (10.6) | 6.2–17.1 |
| | No | 142 | 0 (0.0) | 0.0–3.3 |
| *p*-value | | | 0.001 | |
| Flood plain specific | | | | |
| | Yes | 137 | 15 (11.0) | 6.5–17.7 |
| | No | 5 | 0 (0.0) | 0.0–53.7 |
| *p*-value | | | 0.001 | |

**Table 2.** *Cont.*

| Factors | Category | Frequency | Prevalence (%) n = 15 | 95% CI |
|---------|----------|-----------|----------------------|--------|
| National park | Yes | 142 | 8 (5.6) | 2.6–11.2 |
| | No | 142 | 7 (4.9) | 2.2–10.3 |
| p-value | | | 0.611 | |
| National park specific | | | | |
| | Yes | 8 | 8 (100.0) | 59.8–100.0 |
| | No | 134 | 7 (5.2) | 2.3–10.9 |
| p-value | | | 0.001 | |
| Grazing with wild animals | Yes | 142 | 8 (5.6) | 2.6–11.2 |
| | No | 142 | 7 (4.9) | 2.2–10.3 |
| p-value | | | 0.611 | |
| Grazing with wild animals specific | | | | |
| | Yes | 87 | 8 (9.2) | 4.3–17.8 |
| | No | 55 | 7 (12.7) | 5.7–25.1 |
| p-value | | | 0.019 | |
| | Overall | 15 | 10.6 | 6.2–17.1 |

*n*, number of participants; CI, 95% confidence interval.

### 3.3. Prevalence of NTM in Water

A total of 231 water samples from different drinking water points were collected. Of these, boreholes (a deep hole made in the ground to access water with a hand pump installed on it) were the most common source of water 78.4% (181/231) followed by dam/pond/river sources 14.3% (33/231), and the least common source of water was tap water 7.3% (17/231). The majority of the water samples were collected from rural areas 65.4% (151/231) and the fewest water samples were collected from urban areas 15.6% (36/231).

The overall prevalence of NTM in water was 10.4% (24/231, 95% CI 6.9–15.2). Boreholes reported a comparably higher prevalence of 7.8% (18/231, 95% CI 4.8–12.2), while dam/pond/river sources reported the lowest prevalence of 0.9% (2/231, 95% CI 0.2–3.4) ($p = 0.033$). Tap water had a relatively higher category-specific prevalence of 23.5% (4/17, 95% CI 7.8–50.2) as compared with borehole water, 9.9%% (18/181, 95% CI 6.2–15.5) and dam/pond/river sources, 6.1% (2/33, 95% CI 1.1–21.6) ($p = 0.021$). Chipata District reported a relatively higher prevalence of 8.2% (19/231, 95% CI 5.2–12.7) than Lundazi District, 1.3% (3/231, 95% CI 0.3–4.1) and Namwala District, 0.9% (2/231, 95% CI 0.2–3.4) ($p = 0.016$). Water from rural areas had a high prevalence of NTM contamination, 6.9% (16/231, 95% CI 4.1–11.2) as compared with water from peri-urban and urban areas which had prevalences of 2.2% (5/231, 95% CI 0.8–5.3) and 1.3% (3/231, 95% CI 0.3–4.1), respectively ($p = 0.333$) (Table 3).

**Table 3.** Overall prevalence of NTM infection in water.

| Factors | Category | Frequency | Prevalence (%) n = 24 | 95% CI |
|---------|----------|-----------|----------------------|--------|
| Water source | Borehole | 231 | 18 (7.8) | 4.8–12.2 |
| | Dam/pond/river | 231 | 2 (0.9) | 0.2–3.4 |
| | Tap | 231 | 4 (1.7) | 0.6–4.7 |
| p-value | | | 0.033 | |
| | Borehole | 181 | 18 (9.9) | 6.2–15.5 |
| | Dam/pond/river | 33 | 2 (6.1) | 1.1–21.6 |
| | Tap | 17 | 4 (23.5) | 7.8–50.2 |
| p-value | | | 0.021 | |
| Purpose of water | Domestic | 231 | 19 (8.2) | 5.2–12.7 |
| | Livestock | 231 | 1 (0.4) | 0.0–2.8 |
| | Both | 231 | 4 (1.7) | 0.6–4.7 |
| p-value | | | 0.038 | |

**Table 3.** *Cont.*

| Factors | Category | Frequency | Prevalence (%) $n = 24$ | 95% CI |
|---|---|---|---|---|
| Purpose of water specific | | | | |
| | Domestic | 190 | 19 (10.0) | 6.3–15.4 |
| | Livestock | 17 | 1 (5.9) | 0.3–30.8 |
| | Both | 24 | 4 (16.7) | 5.5–38.2 |
| *p*-value | | | 0.036 | |
| Location | Urban | 231 | 3 (1.3) | 0.3–4.1 |
| | Peri-urban | 231 | 5 (2.2) | 0.8–5.3 |
| | Rural | 231 | 16 (6.9) | 4.1–11.2 |
| *p*-value | | | 0.333 | |
| Location specific | | | | |
| | Urban | 36 | 3 (8.3) | 2.2–23.6 |
| | Peri-urban | 44 | 5 (11.4) | 4.3–25.4 |
| | Rural | 151 | 16 (10.6) | 6.4–16.9 |
| | | | 0.899 | |
| Province | Eastern | 231 | 22 (9.5) | 6.2–14.2 |
| | Southern | 231 | 2 (0.9) | 0.2–3.4 |
| *p*-value | | | 0.018 | |
| Province specific | | | | |
| | Eastern | 213 | 22 (10.3) | 6.7–15.4 |
| | Southern | 18 | 2 (11.1) | 1.9–36.1 |
| *p*-value | | | 0.899 | |
| District | Lundazi | 231 | 3 (1.3) | 0.3–4.1 |
| | Chipata | 231 | 19 (8.2) | 5.2–12.7 |
| | Namwala | 231 | 2 (0.9) | 0.2–3.4 |
| *p*-value | | | 0.016 | |
| District specific | | | | |
| | Lundazi | 64 | 3 (4.7) | 1.2–13.9 |
| | Chipata | 149 | 19 (12.8) | 8.0–19.4 |
| | Namwala | 18 | 2 (11.1) | 1.9–36.1 |
| *p*-value | | | 0.721 | |
| | Overall | 24 | 10.4 | 6.9–15.2 |

*n*, number of participants; CI, 95% confidence interval.

### 3.4. Overall Prevalence of NTM in Humans, Cattle, and Water

The overall prevalence of NTM in humans, cattle, and water was 9.1% (72/794, 95% CI 7.21–11.34). There was no significant difference in prevalence among the three sources, despite cattle presenting a higher prevalence of 10.6% (15/142, 95% CI 6.2–17.1) and humans having the lowest prevalence of 7.8% (33/421, 95% CI 5.5–10.9) (*p* = 0.106) (Table 4).

**Table 4.** Prevalence of NTM in humans, cattle, and water.

| Variable | *n* | Positives | Prevalence (%) | 95% CI |
|---|---|---|---|---|
| Human | 421 | 33 | 7.8 | 5.5–10.9 |
| Cattle | 142 | 15 | 10.6 | 6.2–17.1 |
| Water | 231 | 24 | 10.4 | 6.9–15.2 |
| Overall | 794 | 72 | 9.07 | 7.21–11.34 |
| *p*-value | | | 0.106 | |

*n*, number of isolates; CI, 95% confidence interval.

### 3.5. NTM Species Distribution at the Human–Livestock–Environment Interface

A total of 49 NTM isolates representing 19 species were detected after sequencing. From sputum samples, three MTB species and one Rhodoccus species were also detected, which were removed from further analysis. Overall, the most common isolated species at the interface were *M. avium* complex and *M. fortuitum* at 18.4% (9/49) each. The most isolated NTM species in the different sources were *M. avium* complex in sputum (8/28),

*M. gordonae* in cattle tissues (2/4), and *M. fortuitum* in water (5/17). *M. fortuitum* and *M. gordonae* were isolated from all three sources, while *M. abscessus* was isolated from humans and water (Table 5).

**Table 5.** NTM species distribution at the human–livestock–environment interface.

| | NTM Species | Source | | | Frequency (%) |
|---|---|---|---|---|---|
| | | Human Sputum | Cattle Tissues | Water | |
| 1 | *M. abscessus* | 2 | | 2 | 8.2 |
| 2 | *M. pulveris* | 1 | | | 2.0 |
| 3 | *M. kumamotonense* | 1 | | | 2.0 |
| 4 | *M. rutilum* | 1 | | 2 | 6.1 |
| 5 | *M. smegmatis* | 1 | | | 2.0 |
| 6 | *M. avium* complex | 9 | | | 18.4 |
| 7 | *M. fortuitum* | 3 | 1 | 5 | 18.4 |
| 9 | *M. boenickei* | 1 | 1 | | 4.1 |
| 10 | *M. littorale* | 3 | | | 6.1 |
| 11 | *M. parascrofulaceum* | 1 | | | 2.0 |
| 12 | *M. gordonae* | 1 | 2 | 1 | 8.2 |
| 13 | *M. phocaicum* | | | 4 | 8.2 |
| 14 | *M. mucogenicum* | | | 1 | 2.0 |
| 15 | *M. cosmeticum* | | | 1 | 2.0 |
| 16 | *M. species* | 2 | | | 4.1 |
| 17 | *M. pulveris* | 1 | | | 2.0 |
| 18 | *Coinfection of M. parascrofulaceum* and *M. europaeum* | 1 | | | 2.0 |
| 19 | *M. Senegalence* | | | 1 | 2.0 |
| | **TOTAL** | **28** | **4** | **17** | **100.0** |

NTM, nontuberculous mycobacteria.

## 4. Discussion

In the current study, we aimed to investigate the prevalence and species distribution of NTM at the human–livestock–environment interface in Zambia.

We reported an overall prevalence of NTM isolates in humans, cattle, and water of 9.1%; in addition, we found a prevalence of 7.8% in sputum specimens from patients with presumptive TB, 10.6% in cattle tissues, and 10.4% in water. The isolation of NTM from the different sources implied the possibility of NTM transmission from the environment which is shared between humans and animals at the human–livestock–environment interface [10,33].

The obtained prevalence of 7.8% for humans was similar to those reported by other authors elsewhere, i.e., 7.5% (Kenya), 9.3% (Mali), 8.1% (Tanzania), and 7.5% (Sub-Saharan Africa), and lower than those reported in Cambodia (10.8%), Nigeria (15%), and Ghana (33.2%) [15,19,34–37]. The similarity in the prevalence implies that NTM is becoming a disease of public health importance [26]. Similar studies conducted in Zambia have reported varying prevalences of NTM from 4.6% in presumptive TB patients [38], 11% in patients, and 6% in controls [39], as well as 15.1% based on a country survey [40]. The present study's prevalence rates were comparable to those reported by Chanda et al. (2015) who found that Eastern and Southern provinces had lower prevalence rates of NTM compared with Western and Central provinces which were the highest [40]. These results indicate that NTM continues to be a public health threat, and it also suggests the possibility of NTM-infected patients in the study

being subjected to unnecessarily long treatment with anti-TB drugs [36,41,42]. Such patient mismanagement may negatively impact the health status of individuals and consequently, may cause an additional cost to the health system [40]. Hence, there is a need for physicians to integrate NTM management into current efforts toward the prevention and control of TB. Clinicians should also take into consideration the possibility of NTM infection in patients not responding to TB treatment and also in smear-negative patients with recurrent respiratory infections [42]. In addition, there is also a need to identify species to confirm the clinical significance of the isolated NTM.

Several studies have shown that older females, in general, have been more affected by NTM than males [38,43–46]. To the contrary, our results showed that middle-aged males (4.5%) were more prone to NTM infection than females (3.3%). Our results were consistent with other studies that found males to be the most affected gender with NTM [45,47–49], attributed to the possible susceptibility of males to NTM infection due to higher historical rates of smoking, alcohol abuse, drug abuse, and other respiratory diseases that are more common in males such as chronic obstructive pulmonary disease (COPD) [3,50,51]. In addition, a higher index of suspicion of TB is common in men, and therefore, an increased submission of investigative samples from men may also have contributed to these findings [51].

Nontuberculous mycobacteria are known to affect the elderly over 60 years of age [3,44,46,52] due to failure of the immune system, unfortunately, this is contrary to the findings reported in the current study, as the most commonly affected age group was 31–40 years, followed by the age group between 41 and 50 years. Our results were similar to studies conducted by others who found that NTM had a greater effect on the middle-aged group [15,19,35,36,46]. This age group was among the most active workforce and, considering the nature of the study sites, most of the study participants were farmers. According to a study conducted in Indonesia [50], farmers were found to be more at risk of NTM, since farmers spent a longer time in contact with the environment. This is because NTM are ubiquitous in the environment with the heaviest concentration of the bacteria found in soil and water [5,46,48,53] and the probability of NTM disease increases with the extent of environmental exposure [51]. The other reason for the high prevalence of NTM infection in this age group could be due to high-risk behaviors such as smoking, alcohol abuse, and drug abuse in young people, while such behaviors are not as common among the elderly [51,54]. Therefore, finding a high prevalence of NTM among the most active workforce of the nation causes a threat to the development of the nation and calls for serious awareness and effective management and control of the NTM burden in Zambia.

Furthermore, in the current study, we reported a cattle tissue prevalence of 10.6% which was comparable with the results of other studies reported elsewhere, in India, 10% in cattle and buffalo [55], and in Tanzania [56] and Rwanda, 10.6% and 12.0%, respectively, in cattle [57]. This was however, lower than what has been reported in Ghana (64%) [58] and in Mexico (46.2%) [59] and higher than what has been reported in Uganda (9.1%) [7] and in Tanzania (7.1%) [33]. These differences could be attributed to differences in the, age, breed, animal population, disease status of the animals sampled, differences in sample sizes, different methods of NTM isolation, type of production from where the slaughtered animals originated, level of contact with other animals from different herds, contact with wildlife, competence of the person identifying the tubercle lesion, and lastly, the regional TB incidence differences [60,61].

All the cattle positive for NTM were from Namwala District of the Southern province, this could be because of increased livestock production in the province as compared with Eastern province. According to [62], Namwala District has the highest number of traditionally owned cattle in Zambia. This large number of animals coupled with the practice of free grazing in the Kafue floodplain allows more time for interaction among animals; additionally, overlapping of grazing of this livestock and wildlife in the Kafue floodplain increases the possibility of infection transmission [63,64]. In addition, the open grasslands, dams, and rivers with reeds in Namwala District enhance the formation of

biofilms which increases the multiplication of NTM and while grazing the animals can acquire the infection [32]. The results of the study also showed a higher prevalence of NTM in housed animals. This poses a public health risk as most of these houses/kraals for animals are usually in close proximity with people's houses to deter livestock theft, thus, raising the potential of transmission of the bacteria from animals to humans and vice versa [65]. There is an inverse relationship between the distance of the location of the kraal from people's homes and the risk of acquiring mycobacterial infections in households.

Several studies have shown that bulls are more likely to be infected with mycobacteria, both bovine tuberculosis (bTB) and NTM, than cows [56,66,67]. These studies were in agreement with the results of this study in which the prevalence of NTM was higher in bulls than cows. The reason for this difference in prevalence could be due to differences in the purposes of the animals. Bulls, especially castrated ones, are used as oxen, and hence, they are kept longer and they are also usually in contact with other animals which increases their exposure to infection, unlike cows which are usually kept for breeding and milk production [67]. The other reason for this difference could be that, since NTM is an environmental bacteria, exposure of bulls to dust during ploughing increases their exposure and inhalation of the disease as compared with cows that are not used for ploughing in most communities [5].

Having larger herds increases exposure to mycobacterial infections. This is because large herds tend to travel longer distances looking for larger grazing areas and, in the process, increase the chances of mixing with other animals from a different location. Moreover, the chances of the animals having the same watering and grazing points as wild animals are increased during the dry season, and hence, the risk of contracting mycobacteria infection is increased [60,64,68]. This was in agreement with the results of the current study in which the prevalence of the infection was higher in those animals with a larger herd size of over 50 animals and also in those animals which had a common grazing point with wildlife.

Older cattle over 6 years were found to have a higher prevalence of NTM than the younger animals in this study. This was consistent with studies performed elsewhere which reported increasing exposure to mycobacterial infections increased with age [56,64,66–68], and the longer an animal lives the higher the chances of exposure to mycobacterial infections. This poses a zoonotic potential as, in most areas, the older animals are the ones mostly disposed of for slaughter and humans can get the infection after consuming the animal products without proper processing. Hence, there is a need for concerted efforts from both the veterinary and the medical side to control the disease in humans as well as animals [58,67]. The other reason why older animals had a higher prevalence could be due to immune suppression owing to their older age [56].

In the current study, we reported a prevalence of 10.4% NTM in water samples collected from various drinking water points This prevalence was consistent with what has been reported elsewhere in Uganda (10.39%) [69] and in Iran (10.0%) [70] but lower than that reported in other studies conducted in Italy (72%) [11] Iran (21.4%) [71] Mexico City (16%) [72], and Zambia (15.4%) [32]. These differences could be as a result of differences in sample sizes and different methods of isolation.

The current study obtained a high isolation of NTM from the borehole water source, followed by tap water, and the lowest isolation rate from dam/pond/river sources. These results were similar to those reported previously in Namwala District [32] in which the borehole water source had the highest isolation rate. The high isolation rate could be attributed to a high level of organic matter and soil in borehole water which contributes to mycobacteria flora, and also the piping system used in boreholes may also support the formation of biofilm which favors bacterial growth and multiplication, while treatment with chlorine in tap water, which is usually from a water distribution system, could be lethal to the mycobacteria [32]. To the contrary, Falkinham et al. [6,73,74] reported that there was a high number of NTM in tap water from water distribution systems, since the chlorine administered to water kills off the other competitors for nutrients, allowing the growth of NTM on low concentration of nutrients, and since NTM are thought to be

resistant to chlorine, and thus, they tend to persist in drinking water supply systems and can be transmitted via water and result in PD, disseminated infection, skin/soft tissue infections both in humans and in animals [73]. This could partly explain the higher specific prevalence in tap water as compared with boreholes and dam/pond/river sources of water in the current study.

Furthermore, 19 different NTM species were isolated at the human–livestock–environment interface; *M. avium* complex and *M. fortuitum* were the most commonly isolated species followed by *M. abscessus*, *M. gordonae,* and *M. phocaicum*. Other species isolated at a lower rate included *M. pulveris*, *M. kumamotonense, M. rutilum, M. smegmatis, M. boenickei, M. littorale, M. parascrofulaceum, M. mucogenicum, M. cosmeticum, M. Senegalence,* and a co-infection of *M. parascrofulaceum* and *M. europaeum*. Some of the species were isolated for the first time in Zambia and these included *M. phocaicum*, *M. pulveris*, *M. rutilum*, *M. smegmatis*, *M. boenickei*, *M. littorale,* and *M. europaeum.*

In humans, MAC was the most commonly isolated NTM species which was in agreement with other studies conducted elsewhere [25,37,48,75–78]. MAC is widely distributed in nature, and this increases its chances of spreading and infecting humans [7,79]. It is the major cause of NTMPD, and infection with *M. avium* can have clinical and radiologic presentations indistinguishable from those of TB making it difficult to differentiate and diagnose [25,80].

*M. fortuitum* and *M. gordonae* were isolated from all three sources; the isolation of similar NTM species from humans, animals, and the environment indicate that NTM species from these different sources might be mixing. This may pose a risk of zoonotic transmission of the organism from the environment to humans and animals, as these isolated organisms are potentially pathogenic both to humans and animals [20,33,81].

*M. fortuitum* has been associated with severe cases of wounds and catheters [82]. In addition, *M. gordonae* is a known laboratory and tap water contaminant, though it may be pathogenic in some individuals causing systemic symptoms and may also cause disseminated disease in advanced HIV patients [82]. In cattle, *M. fortuitum* and *M. gordonae* may interfere with a bovine TB diagnosis by eliciting a reaction to purified protein derivative in a bovine skin test, thus, leading to a false positive test result [60]. Therefore, there is a need for a one-health approach in dealing with infections caused by NTM at the interface [33].

*M. abscessus* has been isolated from humans and water. *M. abscessus* is a potentially pathogenic NTM species which is capable of causing more than 30% of pulmonary NTM disease and it is highly resistant to antimicrobials, making treatment of infections by this organism very difficult. In addition, *M. abscessus* can survive in harsh environments such as chlorinated water [77,83,84]. Therefore, isolation of these possible pathogenic organisms in water poses a risk of transmission of these organisms from the environment to humans and animals.

A limitation of this study was that the study sampled presumptive TB patients, hence, this prevalence could not be generalized to the whole population.

## 5. Conclusions

The current study has shown, for the first time in Zambia, simultaneous isolation of NTM in humans, cattle, and water at the human–livestock–environment interface, suggesting the possibility of NTM transmission from the environment (water) to cattle and humans at the interface. *M. avium* complex and *M. fortuitum* were the most commonly isolated species at the interface. *M. fortuitum* and *M. gordonae* were isolated from all three sources, while *M. abscessus* was isolated from humans and water. The isolation of similar NTM species at the human–livestock–environment interface which are potentially pathogenic poses a risk of infection both for humans and animals. Therefore, understanding the disease transmission dynamics at the interface requires a "One Health approach" in dealing with this infection in humans and livestock, and also the need to include water treatment as a way of preventing the disease. Further, there is also a need to incorporate molecular tools such as 16S rRNA and 16S

to 23S rRNA (ITS) sequencing to the diagnosis of NTM in Zambia for species identification, as different NTM species respond to different antimicrobials.

**Author Contributions:** Conceptualization, M.Z., S.M. and M.M.; methodology, M.Z., N.M., S.M. and M.M.; software, M.Z. and H.K.; validation, F.M., R.T. and V.D.; formal analysis, M.Z., M.M. and H.K.; investigation, M.Z., A.N.M., J.N., O.S. and N.M.; resources, M.Z., S.M., N.M. and M.M.; data curation, M.Z., S.A.S. and N.M.; writing—original draft preparation, M.Z.; writing—review and editing, M.Z., H.K., S.M., V.D., F.M., R.T., M.M., S.A.S., A.N.M., J.N., O.S. and N.M.; visualization, M.Z., H.K., R.T. and N.M.; supervision, M.Z., S.M. and M.M.; project administration, M.Z; funding acquisition, M.Z. and M.M. All authors have read and agreed to the published version of the manuscript.

**Funding:** Funding for this study was by the African Centre of Excellence for Infectious Diseases in Humans and Animals in Conjunction with the University of Zambia (ACEIDHA-UNZA), grant number P151847, funded by the World Bank and the APC was funded by the ACEIDHA.

**Institutional Review Board Statement:** This study was conducted in accordance with the Declaration of Helsinki, and approved by the University of Zambia Biomedical Research Ethics Committee (UNZABREC) of the University of Zambia (protocol code 621-2019) for studies involving humans and animals. Thereafter, regulatory approval was granted by the National Health Research Authority (NHRA). Each participant completed a consent form to participate in the study.

**Informed Consent Statement:** Informed consent was obtained from all subjects involved in the study.

**Data Availability Statement:** Data supporting the results in this study can be made available on request from the corresponding author.

**Acknowledgments:** The authors are grateful to the University of Zambia, School of Veterinary Medicine, Department of Disease Control, and to the provincial and district health offices and veterinary officers for Southern and Eastern provinces, as well as the staff at the TDRC TB laboratory for their support during the execution of this study. Additionally, we are thankful to all the patients that participated in the study and all the technical staff and the abattoir staff for their support.

**Conflicts of Interest:** The authors declare no conflict of interest.

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
