# Peer review of "Emergence of Nontuberculous Mycobacteria at the Human–Livestock–Environment Interface in Zambia"

_2036-7481, doi:10.3390/microbiolres14010032_

Round 1

Reviewer 1 Report

INTERESTING MANUSCRIPT. JUST FEW DETAILS :

2.2.3. Water Samples : Hundred and fifty mills (150mls)

IN TABLE 11.0 : AS PARASCROFULACEUM IS SHOWN LINE 11 LINE 18 HAS TO BE MODIFY, INDICATING WHY 2 SPECIES ARE FOUND IN ONE SAMPLE ( COINFECTION IS ONLY INDICATED IN THE DISCUSSION)

« a high prevalence of NTM among the most active workforce of the nation causes a threat

to the development of the nation and calls for serious awareness and effective management

and control of NTM burden in Zambia » . OK, BUT IT WOULD INDICATE THAT THE CONFUSION FACTORS SMOKING, ALCOHOL ABUSE, DRUG ABUSE ARE NOW BETTER CONTROLED THAN PREVIOUSLY IN OLD FARMERS OR.. ARE INCREASING IN YOUNG PEOPLE.

Reviewer 2 Report

Although the article was evaluated as a due diligence, it was found important as it provides up-to-date information about this country.

Major revisions:

In order to be in a one health concept, it must be determined that there is transmission from animal to human. The evidence on this is not clear in this study.

Tables must be combined. Tables are difficult to follow. Therefore, it should be easy to understand.

The authors said that:

The reasons for this increase are not clear, but are thought to be due to increased cultures of the organism,, increase in diseases that confer susceptibility to NTM (e.g chronic obstructive pulmonary disease (COPD) and bronchiectasis and also increased awareness of the disease.

I recommend the authors to revise these sentences with the support of current literature.

REVIEW article

Front. Immunol., 03 March 2020
Sec. Microbial Immunology
Volume 11 - 2020 | https://doi.org/10.3389/fimmu.2020.00303

The increase in research into the epidemiology, diagnostics, and treatment of this once obscure disease stems from the increasing numbers of cases being identified from populations with previously unknown and currently unidentified risk factors (12). Advances in therapeutics in all fields of medicine have seen unexpected NTM disease susceptibilities emerge which pose a challenge in terms of patient care but also provide insight into disease pathology. 

Some minor revisions:

-          There is too much space between words. I marked it in yellow in maintext.

-          Punctuation marks are forgotten. I highlighted in abstract section.

-          Too many commas in sentence. It should be corrected.

Round 2

Reviewer 2 Report

The major problems are:

Is this an appropriate concept for a special issue of the Microbiology Research?

Special issue: Zoonotic Bacteria: Infection, Pathogenesis and Drugs

Although a bacterial species with zoonotic potential has been investigated, zoonotic transmission has not been proven in the article.

There are too many tables. This makes it difficult to follow.

Minor revisions:

Lines 43-45. “The isolation of similar NTM species at the interface which are potentially pathogenic calls for a One health approach in dealing with this infection in humans and livestock and also the need to include water treatment as a way of preventing the disease.”

Last sentence in abstract section - I think it will be better if this sentence is corrected except for the concept of one health. Because in this study, you did not report any infection transmitted from animal to human. It is highly probable that there may be a waterborne transmission to both animals and humans.

Line 61 - Nontuberculous mycobacteria are ubiquitous in the environment with the heaviest ….. “Nontuberculous mycobacteria” please use abbreviated form “NTM”.

Table 11.0.  

Generally, Mycobacterium should be abbreviated as M.

14- Mycobacterium mucogenicum -- Excess line spacing should be deleted.

16- Mycobacterium species which species?

18- “Coninfection of M. parascrofulaceum and M. europaeum” not Coninfection……Coinfection

Author Response

REVIEWER 2 ROUND TWO

The major problems are:

Is this an appropriate concept for a special issue of the Microbiology Research?

Special issue: Zoonotic Bacteria: Infection, Pathogenesis, and Drugs

Although a bacterial species with zoonotic potential has been investigated, zoonotic transmission has not been proven in the article.

Thank you for the comment. The paper was submitted to the journal Microbiology Research, when submitting we did not choose any special issue we just submitted it to the journal, the special issue was just chosen for us which I believe was because of the tittle which had one health concept.

There are too many tables. This makes it difficult to follow.

Thank you for the observation. Tables one to three presenting the background information about the three categories of our samples have been removed.  

Minor revisions:

Lines 43-45. “The isolation of similar NTM species at the interface which are potentially pathogenic calls for a One health approach in dealing with this infection in humans and livestock and also the need to include water treatment as a way of preventing the disease.”

Last sentence in abstract section - I think it will be better if this sentence is corrected except for the concept of one health. Because in this study, you did not report any infection transmitted from animal to human. It is highly probable that there may be a waterborne transmission to both animals and humans.

Thank you for your comment. The sentence has been revised it now reads ‘“The isolation of similar NTM species at the interface which are potentially pathogenic is a public health problem which merits further investigation.

Line 61 - Nontuberculous mycobacteria are ubiquitous in the environment with the heaviest ….. “Nontuberculous mycobacteria” please use abbreviated form “NTM”.

Thank you for the suggestion. The word Nontuberculous mycobacteria has been replaced with the abbreviation NTM.

Table 11.0.  

Generally, Mycobacterium should be abbreviated as M.

Thank you for your observation, the word Mycobacterium has been abbreviated with M.

14- Mycobacterium mucogenicum -- Excess line spacing should be deleted.

Thank you for your observation, the excess line space has been deleted

16- Mycobacterium species which species?

Thank you for the comment. These could not be identified to species level, they were just identified as Mycobacterium species in BLAST analysis

18- “Coninfection of Mparascrofulaceum and Meuropaeum” not Coninfection……Coinfection

Thank you for your observation, the word coninfection has been deleted and the correct word coinfection has added
